# Barriers and opportunities in developing community-based maternal and child health surveillance: A mixed methods study in Depok, Indonesia

Fathimah S. Sigit[iD][1]*, Fitra Yelda[2], Dumilah Ayuningtyas[3], Asri C. Adisasmita[4], Sabarinah Prasetyo[5]

1 Department of Public Health Nutrition, Faculty of Public Health, Universitas Indonesia, Depok, Jawa Barat, Indonesia, 2 Center for Health Research, Faculty of Public Health, Universitas Indonesia, Depok, Jawa Barat, Indonesia, 3 Department of Health Policy and Administration, Faculty of Public Health, Universitas Indonesia, Depok, Jawa Barat, Indonesia, 4 Department of Epidemiology, Faculty of Public Health, Universitas Indonesia, Depok, Jawa Barat, Indonesia, 5 Department of Biostatistics and Population Studies, Faculty of Public Health, Universitas Indonesia, Depok, Jawa Barat, Indonesia

* fathimah10@ui.ac.id

## Abstract

### Background

Comprehensive health surveillance for vulnerable populations, particularly mothers and children, is essential beyond traditional surveys. It may help address gaps in identifying issues occurring outside health facilities or linked to social stigma.

### Methods

This embedded mixed-methods study aimed to identify factors contributing to an effective community-based surveillance system for maternal and child health. Quantitative data on knowledge, attitudes, perceptions, and stigma were collected through interviewer-assisted questionnaires in 300 households. Qualitative insights into barriers and opportunities for detecting, reporting, and monitoring maternal and child health issues were obtained through in-depth interviews and focus group discussions. Participants included public officials, community leaders, medical staff, and social workers. All data were collected across three subdistricts in Depok.

### Results

The household survey revealed that 22.5%, 24.1%, and 15.1% of respondents lacked knowledge of warning signs during pregnancy, childbirth, and newborn care, respectively. Men were less knowledgeable (34.2%, 35.2%, 23.2%) than women (10.7%, 12.9%, 7.0%). Most respondents (98%) supported community-based health monitoring. Thematic analysis unveiled barriers such as the high mobility of migrant families,

**Data availability statement:** Data cannot be shared publicly because of confidentiality issues, as there was some personal information related to the participants. Data are available from the Research and Community Engagement Ethical Committee of the Faculty of Public Health Universitas Indonesia (email: fkmui@ui.ac.id), for researchers who meet the criteria for access to the confidential data.

**Funding:** This study is supported by the Indonesia Endowment Fund for Education (Indonesian: Lembaga Pengelola Dana Pendidikan; LPDP), Grant Number: NKB-696/UN2.RST/HKP.05.00/2021 PRIME (Partnership in Research Indonesia and Melbourne). The funder had no role in study design, data collection and analysis, decision to publish, or preparation of this manuscript. There was no additional external funding received for this study.

**Competing interests:** The authors have declared that no competing interests exist.

inadequate capacity of health volunteers, stigma, delayed healthcare responses, and reluctance among families with middle-to-high socioeconomic status. Alternatively, opportunities included positive community perceptions of surveillance, ongoing community empowerment activities, active roles of health volunteers, potential stakeholder collaboration, and digital communication channels.

## Conclusion

Public health education primarily targeting men is indispensable to enhancing their awareness of maternal and child health issues. Addressing identified barriers and leveraging opportunities could establish a sustainable and well-received community-based surveillance system, crucial for ensuring the health of mothers and children.

## Introduction

Public health surveillance is the systematic collection, collation, and analysis of data for public health purposes. It is an integral part of creating and maintaining healthcare services, as it helps stakeholders detect health problems occurring in society. Information from surveillance may be used to help decision-makers identify modifiable factors in health issues for continued healthcare improvement, or plan actionable public health responses [1–3].

Besides conventional public health surveillance that requires many resources to be conducted, the COVID-19 pandemic has shown us the importance of community-based surveillance, particularly in low- and middle-income countries with limited resources. Community-based surveillance (CBS) is a systematic detection and reporting of events of public health significance within a community by community members, that may serve as an early warning of certain public health issues, which can lead to immediate decision-making of preventive or curative measures [2]. During the COVID-19 pandemic, CBS was used widely worldwide, as community members provided prompt reporting of a probable case of the disease in their community to healthcare providers via a 'real-time' communication network. In the context of COVID-19, CBS helps interrupt transmission through early detection and follow-up response from health authorities of a potential positive case in the community [4]. Unlike the conventional approach in which healthcare facilities usually being the ones gathering data for surveillance, in CBS the community members can report an incident of a (suspected) case in their community, and relay the findings to the integrated reporting and record systems. Hence, it empowers the community to participate in managing public health issues and improving disease preparedness [5,6].

Among public health issues in Indonesia, maternal and child health problems remain one of the leading national public health concerns. Despite steadily decreasing, maternal deaths and other related complications of pregnancy and childbirth are still occurring in Indonesia. For example, chronic energy deficiency is still commonly found in pregnant women, particularly those living in rural areas. Poor pregnancy outcomes such as low birth weight or premature-born babies are also still found.

Previous studies have shown that worse pregnancy outcomes more commonly occur in countries with limited health registration systems, as the lack of data hinders planning and monitoring the progress of public health policies. Thus, in these low-resource settings, community-based surveillance may become an alternative registration system to provide those data [7–9].

Although Indonesia already has an integrated health information and reporting system that was collected by healthcare facilities across the nation, this system is not fully well-established. For example, in rural areas with limited facilities, reports are still commonly collected paper-based, which led to the data not providing a real-time picture of health issues in the population, as it takes time to aggregate the information [1]. This system was also not representative of the underprivileged population who did not access, or had no access to, the formal healthcare facilities, which makes cases (e.g., maternal deaths) that occurred outside healthcare facilities unreported [10]. These gaps may be addressed by community-based surveillance, as a complementary or integral part of the conventional information system [1,9,10].

Nevertheless, despite its benefits and low related costs, CBS is currently not much implemented in Indonesia. Therefore, to help set the foundation for an effective, efficient, and well-received community-based surveillance system, particularly for maternal and child health issues, this study aimed to identify barriers and opportunities related to the surveillance in three sub-districts in Depok, Indonesia.

## Methods

### Study design and population

This is a descriptive cross-sectional study with a mixed methods approach, integrating both quantitative and qualitative data collection and analysis methods. Using an embedded design, the qualitative study provided data supporting the quantitative study. Study recruitment and data collection were performed between 1 November 2022 and 1 October 2023.

The target population in this study were residents living in the city of Depok, Indonesia. Being a supporting neighbour city of the capital Jakarta, Depok is an urban city demographically characterised by its many migrant residents coming from rural areas across Indonesia. Similar to other areas in Indonesia, the primary healthcare service for the population is provided by the Community Health Center, a free-for-public, government-funded healthcare facility.

### Quantitative approach: The interviewer-assisted questionnaire

Quantitative data were collected from 300 households residing in three subdistricts in Depok (Mekarjaya, Curug, and Tirtajaya) with an interviewer-assisted questionnaire. The three subdistricts were purposively selected by the Depok City Health Office, based on their relatively higher prevalence of maternal and children health issues compared to other subdistricts. The sample size was 100 households per subdistrict, calculated based on a 95% confidence level, a 10% precision level, a 50% proportion of events, and a design effect of 1.

The sampling procedure followed a two-stage cluster and random sampling method within each subdistrict to select the households. In the first stage, twenty neighbourhoods/hamlets from each of the three subdistricts were selected as the clusters using the probability proportional to size method. In the second stage, five households per hamlet were randomly selected using a computer-based random generator from the household list of each selected hamlet. Data enumerators visited the households and asked the heads of the households and their spouses to complete the questionnaires.

The cross-sectional questionnaire aimed to assess the level of knowledge, attitudes, and stigmas related to maternal and neonatal health issues and surveillance. These variables were evaluated as understanding the cultural-social norms, perceptions, and attitudes towards health issues is essential to designing a well-accepted relevant surveillance system [8].

### Qualitative approach: In-depth interviews and focus group discussions

To gain comprehensive qualitative insights from multiple perspectives of the community, informants for the in-depth interviews were selected purposively, including community leaders, health volunteers, medical staff at the Community Health

Centre, and stakeholders from the local administrative and health authorities at the subdistrict, district, and city levels. The interviews were done during daytime, working hours, at their offices or places as visited by the data enumerators.

After the interviews, focus group discussions (FGDs) with representatives of the stakeholders and elements of the community were conducted to identify potential solutions or best practices in the community that are applicable to developing a community-based surveillance system. During the FGDs, the moderator guided the discussion using guide topics. The FGDs were held at the University of Indonesia, Faculty of Public Health, and were fully recorded and transcribed by study assistants.

For interviews and FGDs, there was no time limitation per data collection session, but they ended when data saturation was reached, or when no additional information was obtained. The interviews and FGDs were performed in Indonesian language. The involvement of multiple stakeholders/policymakers was aimed to help in planning actionable interventions, as well as reinforcing local policies. The characteristics of the respondents for the in-depth interviews and FGDs were described in S1 Table.

### Data analysis

For the quantitative data, a weighted analysis was performed to account for the sampling design. Data were presented as mean (SD) for continuous variables or proportion (%) for categorical variables. For the qualitative data, all interviews and FGDs sessions were transcribed and re-read, and common themes were identified using the framework approach and qualitative data analysis software (ATLAS.ti Scientific Software Development GmbH). After familiarisation with the data, the main themes were highlighted, and thematic analysis was performed.

### Ethics statement

This study had been reviewed and approved by the Ethics Committee of the Faculty of Public Health, Universitas Indonesia (No.Ket-563/UN2.F10.D11/PPM.00.02/2022). All participants signed written informed consent. No minors were included, and no personal information that could identify individual participants was reported in the present study.

## Results

### Quantitative analysis: Descriptive characteristics of the study population

As shown in **Table 1**, the quantitative study population has an equal proportion of men (50.1%) and women (49.9%). Most respondents who filled in the questionnaires were the heads of the households (43.5%) or their spouses (43.9%). The majority of respondents (87.7%) were married, and approximately half of them (49.9%) were high school graduates. Two-thirds (69.1%) of the respondents never smoked, whereas 24.8% are current smokers.

**Knowledge of warning signs in pregnancy, childbirth, and newborn babies.** When asked about the warning signs that have to be looked into in pregnant mothers, bleeding/haemorrhage (40.5%) and hypertension (18.6%) were the most chosen options from the answers. A quarter (22.5%) of participants responded that they did not know the warning signs. The proportion of respondents who answered that they did not know the warning signs was disproportionately higher in men (34.2%) than women (10.7%) (see **Table 2**).

In regard to warning signs during childbirth, around 25.1% of respondents chose premature rupture of the membrane, 42.7% answered massive bleeding during and after delivery, and 10.9% answered prolonged contraction or prolonged labour. A quarter respondents (24.1%) answered that they did not know the warning signs during childbirth, with again a disproportionate proportion between men (35.2%) and women (12.9%). When asked about warning signs in newborn babies, the most chosen answers are jaundice (54%), babies not crying (19.3%), and small born or low birth weight babies (17.6%). Around 15.1% answered that they did not know the warning signs in newborn babies, and this is more so in male (23.2%) than female (7.0%) respondents (see **Table 2**).

**Table 1. Descriptive characteristics of study participants [quantitative survey; n=601; 50% men].**

| Characteristics | Men | | Women | | Total | |
|---|---|---|---|---|---|---|
| | n | % | n | % | n | % |
| Relationship with Head of Household | | | | | | |
| Head of household | 247 | 81.9 | 15 | 4.9 | 261 | 43.5 |
| Spouse of the head of household | 2 | 0.6 | 262 | 87.5 | 264 | 43.9 |
| Son/Daughter | 36 | 12.1 | 12 | 3.9 | 48 | 8 |
| Other (Parent/Brother/Sister/In-Law, etc) | 16 | 5.4 | 11 | 3.7 | 28 | 4.6 |
| Marital Status | | | | | | |
| Single/Unmarried | 36 | 12 | 13 | 4.3 | 49 | 8.2 |
| Married | 257 | 85.2 | 270 | 90.2 | 527 | 87.7 |
| Divorced/Separated | 8 | 2.8 | 17 | 5.5 | 25 | 4.1 |
| Education | | | | | | |
| <6 years of education | 1 | 0.2 | 5 | 1.8 | 6 | 1 |
| 6–12 years of primary education | 40 | 13.4 | 57 | 19.2 | 97 | 16.3 |
| 12 years of education (high school graduate) | 162 | 53.6 | 139 | 46.3 | 300 | 49.9 |
| College/University | 99 | 32.8 | 99 | 32.8 | 197 | 32.8 |
| Occupation | | | | | | |
| Housewife/Stay-at-home mother | 0 | 0 | 201 | 67 | 201 | 33.4 |
| Civil servant/government employee | 19 | 6.2 | 7 | 2.4 | 26 | 4.2 |
| Private employee | 66 | 22 | 17 | 5.7 | 83 | 13.9 |
| Farmer/Fisher | 1 | 0.4 | 0 | 0 | 1 | 0.2 |
| Entrepreneur/Merchant | 54 | 17.9 | 32 | 10.6 | 86 | 14.3 |
| Self-employed | 42 | 14.2 | 6 | 1.8 | 48 | 8 |
| Student | 14 | 4.5 | 8 | 2.8 | 22 | 3.6 |
| Unemployed | 27 | 8.9 | 4 | 1.4 | 31 | 5.2 |
| Retired | 41 | 13.7 | 8 | 2.8 | 50 | 8.2 |
| Other | 37 | 12.2 | 17 | 5.5 | 53 | 9 |
| Smoking Status | | | | | | |
| Never smoke | 118 | 39.3 | 297 | 99 | 415 | 69.1 |
| Smoking within this month | 147 | 48.7 | 2 | 0.8 | 149 | 24.8 |
| Smoking within this year | 4 | 1.4 | 0 | 0 | 4 | 0.7 |
| Smoking last year or earlier | 32 | 10.6 | 1 | 0.2 | 32 | 5.4 |

**Attitudes on maternal and child health issues.** The majority of respondents (87.1%) perceive that health monitoring is essential and should be made compulsory. Around 72.9% of respondents agreed that counselling to maintain mothers' and children's health is a duty of health volunteers, and 86.1% agreed that childbirth should be assisted by trained medical professionals, such as midwives and doctors. The majority (82.4%) disagree that pregnancy, delivery, and child growth can happen well if local myths are followed, and 59.9% disagree if childbirth is assisted by non-medical professionals (e.g., shamans) (see S2 Table). When asked about their approval of pregnant mothers, births, and newborn babies in their community being monitored, the majority of (98%) respondents said that they would approve such monitoring to be conducted (see S3 Table).

**Perceptions towards case reporting and healthcare facilities.** Approximately 82% of respondents did not have difficulties reporting health issues related to pregnant mothers, childbirth, or sick newborns. The majority (>82%) also did not have problems obtaining healthcare services (see S4 Table). If issues related to pregnant

**Table 2. Quantitative analysis: Respondent's knowledge of maternal and child health.**

| | Men (n = 301) | | Women (n = 300) | | Total | | p-value |
|---|---|---|---|---|---|---|---|
| | n | % | n | % | n | % | |
| **In your opinion, what are the warning signs of pregnancy?** | | | | | | | |
| Prolonged contraction | 29 | 9.6 | 26 | 8.6 | 55 | 9.1 | 0.778 |
| Bleeding | 106 | 35 | 138 | 46.1 | 244 | 40.5 | 0.008 |
| High fever | 26 | 8.7 | 24 | 7.9 | 50 | 8.3 | 0.883 |
| Convulsions | 3 | 1 | 9 | 2.9 | 12 | 1.9 | 0.089 |
| Fetal malpresentation/malposition | 10 | 3.2 | 5 | 1.5 | 14 | 2.4 | 0.296 |
| Leg edema | 13 | 4.3 | 33 | 11.1 | 46 | 7.7 | 0.002 |
| Loss of consciousness/fainting | 7 | 2.3 | 6 | 2.1 | 13 | 2.2 | 1 |
| Difficulty in breathing | 8 | 2.7 | 25 | 8.4 | 33 | 5.6 | 0.002 |
| Fatigue | 16 | 5.2 | 30 | 10.1 | 46 | 7.6 | 0.032 |
| Hypertension | 32 | 10.5 | 80 | 26.7 | 112 | 18.6 | 0.000 |
| Early loss of amniotic fluid | 18 | 5.9 | 30 | 10.1 | 48 | 8 | 0.072 |
| Decreased fetal movements | 17 | 5.6 | 22 | 7.2 | 38 | 6.4 | 0.414 |
| Persistent vomiting and loss of appetite | 15 | 4.9 | 35 | 11.5 | 49 | 8.2 | 0.003 |
| Other | 66 | 21.8 | 104 | 34.6 | 169 | 28.2 | 0.001 |
| I don't know | 103 | 34.2 | 32 | 10.7 | 135 | 22.5 | 0.000 |
| **In your opinion, what are the warning signs during childbirth?** | | | | | | | |
| Early rupture of the amniotic sac | 58 | 19.1 | 93 | 31.2 | 151 | 25.1 | 0.001 |
| Massive bleeding during and after delivery | 101 | 33.7 | 155 | 51.8 | 257 | 42.7 | 0.000 |
| High fever | 1 | 0.4 | 6 | 2.2 | 8 | 1.3 | 0.068 |
| Prolonged contraction/labor | 29 | 9.7 | 36 | 12.1 | 66 | 10.9 | 0.36 |
| Loss of consciousness/fainting | 7 | 2.4 | 10 | 3.3 | 17 | 2.9 | 0.474 |
| Convulsions | 6 | 2 | 7 | 2.4 | 13 | 2.2 | 0.788 |
| Retention of placenta | 0 | 0 | 0 | 0 | 0 | 0 | – |
| Intrauterine fetal death | 6 | 2.1 | 7 | 2.2 | 13 | 2.1 | 0.788 |
| Hypertension | 37 | 12.4 | 94 | 31.4 | 131 | 21.9 | 0.000 |
| Greenish and foul-smelled amniotic fluid | 5 | 1.5 | 8 | 2.8 | 13 | 2.2 | 0.417 |
| Mother's anxiety or intense pain | 9 | 3.2 | 6 | 2 | 15 | 2.6 | 0.603 |
| Mother's inability to push during labor | 13 | 4.2 | 12 | 3.9 | 24 | 4 | 1.000 |
| Fetal entanglement in the umbilical cord | 19 | 6.3 | 12 | 3.9 | 31 | 5.1 | 0.268 |
| Other | 27 | 9 | 58 | 19.5 | 85 | 14.2 | 0.000 |
| I don't know | 106 | 35.2 | 39 | 12.9 | 145 | 24.1 | 0.000 |
| **In your opinion, what are the warning signs in postpartum mothers?** | | | | | | | |
| Massive bleeding | 84 | 27.7 | 155 | 51.7 | 238 | 39.7 | 0.000 |
| Loss of consciousness/fainting | 2 | 0.8 | 9 | 3.1 | 12 | 1.9 | 0.036 |
| Convulsions | 2 | 0.8 | 4 | 1.4 | 6 | 1.1 | 0.450 |
| High Fever | 15 | 5.1 | 18 | 6.2 | 34 | 5.6 | 0.596 |
| Foul-smelled vaginal mucus | 4 | 1.2 | 6 | 1.9 | 9 | 1.6 | 0.544 |
| Breast pain | 1 | 0.4 | 13 | 4.5 | 15 | 2.4 | 0.001 |
| Sadness or depressed feeling | 9 | 2.9 | 25 | 8.3 | 34 | 5.6 | 0.005 |
| Other | 38 | 12.7 | 73 | 24.5 | 112 | 18.6 | 0.000 |
| I don't know | 170 | 56.6 | 75 | 25 | 245 | 40.8 | 0.000 |
| **In your opinion, what are the warning signs in newborn infants?** | | | | | | | |
| Pale-colored feces | 1 | 0.4 | 1 | 0.4 | 2 | 0.4 | 1.000 |
| High fever | 38 | 12.5 | 63 | 21 | 101 | 16.7 | 0.006 |

*(Continued)*

**Table 2.** (Continued)

| | Men (n=301) | | Women (n=300) | | Total | | p-value |
|---|---|---|---|---|---|---|---|
| | n | % | n | % | n | % | |
| Diarrhea | 4 | 1.4 | 16 | 5.2 | 20 | 3.3 | 0.006 |
| Vomiting | 2 | 0.6 | 5 | 1.6 | 6 | 1.1 | 0.285 |
| Jaundiced eyes and skin | 125 | 41.5 | 200 | 66.6 | 325 | 54 | 0.000 |
| Low body temperature/hypothermia | 4 | 1.4 | 1 | 0.2 | 5 | 0.8 | 0.373 |
| Skin appeared blue | 20 | 6.6 | 36 | 11.9 | 56 | 9.3 | 0.025 |
| Small-born/low birth weight | 47 | 15.7 | 58 | 19.5 | 106 | 17.6 | 0.238 |
| Difficulty breathing | 45 | 15 | 50 | 16.8 | 96 | 15.9 | 0.577 |
| Not crying | 55 | 18.2 | 61 | 20.5 | 116 | 19.3 | 0.536 |
| Grunting | 4 | 1.2 | 1 | 0.4 | 5 | 0.8 | 0.373 |
| Convulsions | 15 | 5.1 | 31 | 10.3 | 46 | 7.7 | 0.014 |
| Unwilling to be fed | 30 | 9.8 | 35 | 11.6 | 64 | 10.7 | 0.514 |
| Redness, foul odor, or pus in the umbilical cord | 4 | 1.2 | 12 | 4.1 | 16 | 2.7 | 0.045 |
| Other | 43 | 14.4 | 49 | 16.4 | 93 | 15.4 | 0.498 |
| I don't know | 70 | 23.2 | 21 | 7 | 91 | 15.1 | 0.000 |

mothers, childbirth, postpartum mothers, and newborn babies occur in the community, respondents believe that they should report to the Community Health Center (43.0–45.2%), nearby hospital (43.5–48.0%), clinics (20.9–22.4%), social worker (10.2–13.0%), health professionals (11.6–12.9%), and community leaders (11.0%−14.1%) (see S5 Table).

**Qualitative analysis: Barriers to establishing a community-based surveillance system for maternal and child health [Table 3]**

1. *The High Mobility of Residents*

The biggest identified challenge is the high mobility of incoming and outgoing migrants from rural areas all over the country who reside temporarily in Depok to seek jobs in the city or the capital, Jakarta. These migrants bring along their families, including their children, who often have undernutrition problems (stunting or wasting). Despite some intervention programs having been carried out to prevent stunting, as the influx exceeds the outflow of migrant families, the number of children with stunting always appears to be high in the city.

This theme emerges repeatedly and consistently across various stakeholders, including district and subdistrict public officials, Community Health Centre medical staff, and local health volunteers, as illustrated in the following direct quotes. Complete statements are available in Table 3.

*"Many incoming migrants, mostly those who rent houses temporarily, are sometimes unnoticed and unreported to the head of the neighborhood … Thus, they become unregistered at the Integrated Health Service Post." (Subdistrict Official)*

2. *Inadequate Capacity of the Health Volunteers*

Another challenge was the inadequate capacity of the health volunteers, which hindered the delivery of surveillance. For example, as health volunteers are generally older, many are not digitally adept at utilising online reporting tools. Also, it was observed that the health volunteers often did not perform anthropometric measurements (weight and

**Table 3. Barriers and opportunities in establishing community-based surveillance on maternal and child health [qualitative analysis].**

| | | Respondents | Statements from Interviews/Focus Group Discussions |
|---|---|---|---|
| **Barriers** | High mobility of incoming and outgoing migrant workers with their children | Subdistrict Official | "Many incoming migrants, mostly those who rent houses temporarily, are sometimes unnoticed and unreported to the head of the neighbourhood. Delayed care for migrant pregnant women and children sometimes happens because of this, as they do not report themselves to the local officials, community leaders, or social health workers. Thus, they become unregistered at the Integrated Health Service Post. I think the head of the villages should go door-to-door to the rented houses to monitor." |
| | Lack of capacity of the health volunteers | Medical Staff from the Community Health Center | "Not all volunteers are digitally adept, particularly the older ones. That is our difficulty when introducing the online surveillance system with Google Forms, as only the young volunteers can do the report via Google Forms." <br> "Another challenge that we met is human error in stunting measurement. We sometimes observed that the health volunteers did not weigh the children in a correct manner, despite training had been provided to them so that they could measure the weight and height of the children correctly and accurately." |
| | Stigma related to sensitive maternal and child health issues | Health Volunteer | "What we often find is that the parents are reluctant if health issues of their children are known by other people, especially those issues that the public sees as a shame, for example, if their children are malnourished or stunting." |
| | Late response from healthcare providers | Head of Village/ Neighbourhood | "If community-based surveillance is about to be implemented, strengthening intersectoral coordination is extremely needed, starting from the village level until the subdistrict, district, and Community Health Center, to ensure follow-up from community reporting is handled promptly. Nowadays, the coordination is still lacking, and the delayed response from the relevant agencies leads to the community becoming reluctant to be involved in the subsequent government programs." |
| | Unwillingness to participate in families with middle-high socioeconomic status | Health Volunteers | "There is a challenge in reaching the residents who live in gated communities, as they are generally very difficult to be visited. We often got rejected when we house visited because they said that they were very busy and always working. They are also, in general, more reserved or less sociable." <br> "Residents with middle-high income socioeconomic status who live in gated communities are generally less involved in community activities. Economically, they have the means to access private healthcare services, and every time health volunteers or staff from the Community Health Center come to their house, they are always difficult to be met. We have to repeat the visit multiple times." |
| **Opportunities** | Community's positive perceptions of surveillance | Head of Village/ Neighbourhood | "In general, the community is open to community-based surveillance. In many instances, community members even have a high awareness of self-reporting their condition, as they know it is needed to make regulations related to healthcare services." |
| | Existing community empowerment activities | District Official, Health Volunteers | "Currently, the 'Blessed Friday' program is regularly conducted in all villages. There, food products such as vegetables, meat, rice, and other things are given to those in need for free." <br> "In the Qurán recitation routine events, the religious leaders and health volunteers always remind the pregnant women and small children to visit POSYANDU and the elderly to go to the POSBINDU."* <br> "If there is a person who is ill, others chip in to help, to visit the ill person, and jointly collect money to give a little financial help to the ill if they are hospitalized. This activity happens often and easily because we often meet each other during the weekly Qurán recitations." |
| | Active role of health volunteers | Subdistrict Official, Community Member | "The health volunteers are completely social-based, with no salary at all, nothing. The city's health office often tells us they have budget limitations to conduct community activities, but the volunteers are OK with that. I greatly respect the volunteers as they are truly social and never ask about transportation costs when doing their activities." <br> "Despite all limitations, the health volunteers always actively do their activities every Friday. They conduct house visits, or what we call "sweeping", in their neighbourhoods to identify if there are new pregnant women in the community. They also always diligently remind pregnant women to check their condition at the Community Health Center, or mothers with small children to visit the POSYANDU." |

*(Continued)*

**Table 3.** (Continued)

| | | Respondents | Statements from Interviews/Focus Group Discussions |
|---|---|---|---|
| | Potential collaboration between stakeholders and community organisations | Medical Staff from the Community Health Center, Health Volunteers | "The 'Blessed Friday' program may also invite charity organizations or private industries via their CSR to participate as donors in providing free foods for those in need in every village. The health office or the city's mayor can help coordinate to involve these donors". "To be able to implement a digital-based surveillance system in the community widely, we will need the involvement from the youth organization as they have bright ideas and are smart. However, many of them have little time as most are still in school or have a day job." "Another institution that is very important to be involved is the academics from universities. For example, the residents are very excited and enthusiastic every time students from Universitas Indonesia come to provide public health education to the community." |
| | Digital communication channels | District Official, Head of Village/ Neighbourhood | "The communication between health volunteers and the Community Health Center is going well. To communicate with each other, there is a WhatsApp group between all volunteers in the district and subdistricts with the medical staff at the Community Health Center. If there is an emergency, the volunteers can contact the Community Health Center immediately to obtain help." "We have a WhatsApp group with all heads of households in the village for means of coordination. For example, during COVID, if there is news of a community member testing positive, the news is shared so everybody directly knows." |

*Abbreviation: POSYANDU (*Pos Pelayanan Terpadu*; The Integrated Health Service Post). POSBINDU (*Pos Pembinaan Terpadu; The Integrated Guidance Post for Noncommunicable Diseases*).

height) in a correct manner to the children at the local 'Integrated Health Service Post' (Indonesian: Posyandu), which is an activity organised regularly by health volunteers in the community to monitor the health of pregnant mothers and children, supervised by the village midwives. This measurement error led to inaccurate reporting of the children's nutritional status.

> "We sometimes observed that the health volunteers did not weigh the children in a correct manner …" (Community Health Center Medical Staff)

### 3. Stigma Related to Sensitive Maternal and Child Health Issues

Sometimes families or community members avoid surveillance or house visits by health volunteers, especially in cases the public sees as shame, such as when mothers are pregnant out of marriage. The avoidance was also shown by their hesitancy to have their conditions checked or examined by health professionals.

> "What we often find is that the parents are reluctant if health issues of their children are known by other people, especially those issues that the public sees as a shame …" (Health Volunteers)

### 4. Late Response from Healthcare Providers

Another potential barrier was the sometimes-late response from healthcare providers after a case was reported. This sometimes makes community members less eager to report when other new issues are found in their community, as they thought their report was meaningless.

> "… The delayed response from the relevant agencies leads to the community becoming reluctant to be involved in the subsequent government programs." (Head of Village)

5. *Unwillingness to Participate in Families with Middle-High Socioeconomic Status*

In addition, sometimes families reject visits from health volunteers simply because they do not know the enumerator/visitor or have no time to be surveyed. This was particularly often in higher-income neighbourhood clusters with middle-high socioeconomic status families.

*"There is a challenge in reaching the residents who live in gated communities, as they are generally very difficult to be visited. We often got rejected when we house visited …" (Health Volunteers)*

**Qualitative analysis: Opportunities in establishing the community-based surveillance [Table 3]**

1.  *The Community's Positive Perceptions on Surveillance*

Most residents perceived that detecting, recording, and reporting certain medical conditions were necessary for health monitoring. There was, in fact, already a reporting mechanism from the community to the Community Health Centre. This was done mainly by health volunteers conducting house visits to the community members who had a suspected case, or the community members informed the health volunteers when they found a suspected case, and the health volunteers relayed the information to the Community Health Centre. The medical staff at the Community Health Centre then verify the instances by checking the health conditions of the individuals with the suspected case.

*"In general, the community is open to community-based surveillance. In many instances, community members even have a high awareness of self-reporting their condition …" (Head of Village)*

2.  *The Existing Community Empowerment Activities*

A best practice that has been organised routinely by community members to their community is the 'Blessed Friday' program, in which in this program, nutritious food products (raw or ready-to-eat) are distributed to families in need every Friday of the week. These foods are prepared by the community for families with socioeconomic disadvantages, and those with stunted children are given priority as targets for this program. Besides the 'Blessed Friday' program, another best practice is the weekly Qurán recitation events, in which in this mass congregation, the community/religious leaders often remind the public to check their condition at the Community Health Center and help those who are ill.

*"Currently, the 'Blessed Friday' program is regularly conducted in all villages. There, food products such as vegetables, meat, rice, and other things are given to those in need for free." (District Official)*

3.  *The Active Role of Health Volunteers*

Health volunteers play a major role in identifying health issues in the community. One of their primary roles is to organise the routine Integrated Health Service Post activities ('Posyandu'), during which the general health of pregnant mothers and children's growth is monitored. The volunteers then report if any high-risk pregnancies are detected, as well as the anthropometric measurement data of the children, to the Community Health Center. In cases of high-risk pregnancies, the volunteers also routinely visit the mothers at home to monitor their general health status.

*"… They (the health volunteers) conduct house visits, or what we call "sweeping", in the neighborhoods to identify if there are new pregnant women in the community." (Community Member)*

4. *Potential Collaboration between Stakeholders and Community Organisations*

Monthly coordination meetings between health volunteers, district and subdistrict officials, and medical staff of the community health centre are regularly held to evaluate and solve health issues in the community. However, they believe that more stakeholders should be invited or involved in the coordination meetings, such as private healthcare services, academic institutions, local (para)military force, youth organisations, and the office of communication and informatics, as they have distinguished roles that each contribute to better surveillance system.

For example, involving private hospitals and midwife practices in the area may strengthen the referral network between healthcare facilities. Academic institutions may aid in providing health information sessions regarding a particular issue in the community, or help detect health issues and suggest relevant potential interventions by doing research. The local (para)military officers may be involved in surveillance activities (e.g., household visits), as the community tend to consider personnel 'in uniform' more seriously. Youth organisations can be asked for their ideas as they are generally more well-educated and innovative, and may explore new funding opportunities to finance the community-based surveillance system. Meanwhile, the Information Office (Diskominfo) can help in conveying public health messages or reducing stigmas of certain sensitive issues.

*"The 'Blessed Friday' program may also invite charity organizations or private industries via their CSR to participate as donors in providing free foods for those in need in every village. The health office or the city's mayor can help coordinate to involve these donors". (Community Health Center Medical staff)*

5. *Digital Communication Channels*

Developing a user-friendly digital tool to record cases is potentially welcomed by the community, and maintenance should be prepared. In fact, in the subdistrict of Tirtajaya, there is already an online Google sheet that can be filled by households and health volunteers whose data is sent directly to the Community Health Centre. This online spreadsheet is called the 'Healthy Family Card', in which households independently report their health issues to the tool.

Besides this online spreadsheet, digital mobile applications, such as WhatsApp groups, are also seen as efficient coordination channels between stakeholders (community health centre staff and the district and subdistrict officials), as well as information source for the community during the Covid-19 pandemic. For example, there is a specific WhatsApp group dedicated to pregnant mothers in the community, in which in the group the health volunteers routinely provide information on maintaining a healthy pregnancy. Social media such as Instagram and TikTok are also seen as communication channels that are particularly relevant and preferred by the younger generation of mothers.

*"To communicate with each other, there is a WhatsApp group between all volunteers in the district and subdistricts with the medical staff at the Community Health Center. If there is an emergency, the volunteers can contact the Community Health Center immediately to obtain help." (District Official)*

## Discussion

This study revealed that around 15.1%−24.1% of the study population appeared to have insufficient knowledge of maternal and child health issues, and this is disproportionately higher in men (23.2%−35.2%) than women (7.0%−12.9%). Most respondents (98%) approved the monitoring of pregnant mothers, childbirth, and newborn babies, and local stakeholders perceived community-based surveillance as an important method to collect complementary data on health events, alongside the conventional surveillance system.

Several opportunities and barriers were observed in relation to the implementation of community-based surveillance. The barriers were the high mobility of incoming and outgoing migrant workers with their children, the inadequate capacity of the health volunteers, stigma related to sensitive maternal and child health issues, delayed response from healthcare providers, and unwillingness to participate among families with middle-high socioeconomic status. On the other hand, the opportunities were the community's positive perceptions of surveillance, the existing community empowerment activities, the active role of health volunteers, potential collaboration between stakeholders and community organisations, and digital communication channels.

Comparing our quantitative results with other studies, our observation that men had less knowledge of warning signs in pregnancy, childbirth, and newborn care was consistent with a study from Mersha (2019), which reported that men were often less aware of obstetric danger signs and were less involved in birth preparedness and complication readiness [11]. This problem is potentially detrimental, as husbands often act as key decision-makers in the household, and would determine whether the mothers and the newborns can obtain healthcare services during critical periods [10]. For example, Rahman et al. (2018) found that pregnant women who were accompanied by their husbands to antenatal care were more likely to receive care from medically trained provider [12]. Similarly, Kebede et al. (2022) also showed that low paternal involvement in maternal and child health services was linked to an increased risk of postpartum depression among mothers [13].

Comparing our qualitative findings to previous studies, our observation that high population mobility is a barrier to employing community-based surveillance also echo prior research. Incoming migrant families often reside temporarily in rented housing and remain unregistered in villages records, making them unreachable by health volunteers and ineligible for government-subsidized health insurance. This is particularly concerning, as studies have shown that children from migrant families are at higher risk of stunting. Rogeaux (2022) found that migrant children in urban areas had lower weight gain, particularly if born before or during migration [14]. Sharma (2023) attributed this to financial hardship, inadequate feeding practices, poor sanitation, and maternal anemia [15]. Khan (2020) further noted that 'urban poor' populations with unfavorable social determinants of health (SDoH) are less likely to access healthcare services [16], exacerbating health disparities among migrant communities.

Our finding was also coherent with previous studies, which observed that stigma remains a barrier to accessing public healthcare services [17,18], leading to decreased quality and continuity of care [19]. For instance, Kim (2019) reported that stigma towards unmarried mothers caused them to hide their pregnancy and avoided antenatal care [20]. Mukaliburt (2022) similarly observed that stigmatization toward tribal ethnic minorities (in Thailand) led to poorer maternal health outcomes, with some facilities even refusing service [18].

Another minor barrier was the delayed response from healthcare providers to cases reported through community-based surveillance. However, since our data were collected near the end of the COVID-19 pandemic, this delay may reflect the overburdened healthcare system during that period. Czeisler (2020) noted that approximately 12% of routine medical care, as well as 32% of emergency care, were delayed during the pandemic [21]. Owusu et al. (2023) also reported that delayed case management was a common challenge faced by community-based surveillance volunteers [22].

Interestingly, we observed that households with higher socioeconomic status, or those residing in gated communities, were less likely to participate in community-based surveillance, emphasizing the need for more adaptive strategies to engage affluent populations. This contrasts with findings from several Western countries, where individuals with higher education and income levels are typically more responsive to both web-based and in-person surveys [23–25]. For example, Roberts et al. (2020) and Jang (2019) reported that in the United States, survey response rates were particularly low among populations with lower education and socioeconomic status [23,24]. Van Loon (2003) highlighted that such non-response can bias prevalence estimates, affecting the validity of surveillance data [25].

Besides the identified barriers, a common theme that emerged as a prominent opportunity for strengthening community-based surveillance is the potential use of digital communication channels. This aligns with prior studies which observed that

digital applications or electronic-based reporting systems improve the acceptability of surveillance in the community. Diese (2018) observed that mobile phones were widely perceived by the community as accessible and practical tools for relaying information regarding health events quickly [26]. Incorporating digital reporting by community members may also provide real-time data with low utilisation costs. Verity et al. (2022) emphasized that timely detection of disease through surveillance can lead to swift communal action and facilitate rapid response from health authorities [27].

Another particular theme that emerged as both an opportunity and a challenge is the role of health volunteers, or community health workers. It is well established that health volunteers have pivotal roles in the community, and this was clearly seen during the COVID pandemic when there was a massive shortage of trained medical professionals, particularly in underserved or rural settings [28,29]. In Thailand, for example, Kaweenuttayanon et al. (2021) reported that during the COVID-19 pandemic, trained village health volunteers played key roles in cases identification, quarantine monitoring, and referral systems [28]. In Somalia, community health workers acted as first responders and identified a third (32.7%) of cases and contacts through community-based surveillance system, particularly in areas distant from healthcare facilities [29]. These successes were possible, as noted by Sakeah et al. (2021), largely driven by the health volunteers' ability to lead community mobilization and support health promotion efforts [30].

Nevertheless, despite their valuable contributions, we observed that systematic errors in the work of health volunteers, such as inaccurate measurements of infant weight and height, can lead to misidentifications of health issues [22,31,32]. This highlights the urgent need for continuous capacity building to ensure the effectiveness and sustainability of health volunteer programs. Nontapet (2022) and Owusu (2023) reported that community health workers perceived regular training as critical for effective disease identification and health promotion [22,31]. Yet, as noted by Bezbaruah (2021), such training is not consistently provided [33]. Sunguya (2017) and Bezbaruah (2021) also highlighted that the lack of essential equipment, such as rapid test kits and basic medications, often constrained the volunteers' ability to perform their roles effectively [32,33].

Furthermore, while altruism and a sense of duty to the community were the biggest motivation for becoming a health volunteer (Dil, 2012), and most do not expect to be paid, Sakeah (2021) found that they still desired that their work be recognized and appreciated by means other than money, such as uniforms and certificates [30,34]. As also suggested by Sunguya (2017), Nontapet (2022), and Sakeah (2021), if the budget allows, small monetary incentives such as transport allowances could also support their fieldwork and enhance program sustainability, particularly in reaching remote areas [30–32].

## Strengths and limitations of the study

This study employs a robust mixed-methods approach, integrating quantitative surveys with qualitative insights to provide a comprehensive understanding of community-based maternal and child health surveillance. The large sample size (300 households) and diverse stakeholder insights enhance representativeness and real-world applicability, capturing on-the-ground challenges and best practices from multiple levels of community and healthcare stakeholders, making the findings highly relevant for policy and intervention planning.

However, several limitations exist. The cross-sectional design restricts the ability to infer causal relationships between identified factors and surveillance effectiveness. Additionally, self-reported data from the household survey may be subject to recall or social desirability bias, particularly on sensitive topics such as stigma and healthcare-seeking behaviors. Although the quantitative survey included community members (heads of households) involved in community-based surveillance, the qualitative analysis has yet to incorporate perspectives from pregnant women, mothers with infants, or men as husbands. Future studies are encouraged to include these groups in qualitative interviews to better capture end-user perspectives. Lastly, while the study covers three subdistricts in Depok, its generalizability to other regions in Indonesia may be limited due to variations in health system infrastructure and socio-cultural factors. Future research should consider longitudinal studies and broader geographic coverage to strengthen applicability.

## Public health implications

This study highlights several actionable strategies to strengthen maternal and child health surveillance and service delivery. First, targeted maternal and child health education, particularly for men on obstetric danger signs, alongside efforts to foster paternal involvement, should be promoted, as husbands play a crucial role in household decision-making and access to healthcare services. Second, to prevent under-surveillance problems among migrant populations, gated communities, and stigmatized cases, tailored approaches could work, including: (1) proactive outreach in densely populated rental housing clusters and informal settlements to better reach migrant households; (2) cooperation with resident associations and private clinics to improve surveillance participation among higher socioeconomic groups in gated communities; and (3) public campaigns to reduce stigma and normalize health-seeking behaviour to better to capture sensitive cases, e.g., unwanted pregnancies. Third, leveraging user-friendly digital tools such as mobile-phone-based data reporting systems may help increase participation in community-based surveillance, including in the three aforementioned groups. A pilot initiative integrating digital tools into community-based surveillance is recommended as the next step, with this study serving as a benchmark for future evaluations. Lastly, strengthening the role, capacity, and support for health volunteers is essential, and this can be achieved by providing regular training and basic equipment, genuine recognition, and small incentives (transport allowances) to sustain volunteer performance. These strategies can be implemented through intersectoral collaboration across the health, social welfare, education, and civil registration sectors.

## Conclusion

This study highlights the barriers and opportunities in establishing a community-based surveillance system for maternal and child health in Depok, Indonesia. We observed that while most community members support health monitoring, knowledge gaps—particularly among men—persist, and logistical challenges such as high population mobility, health volunteer capacity limitations, and social stigma hinder effective implementation. However, strong community engagement, existing empowerment programs, digital communication channels, and stakeholder collaboration offer promising opportunities. Addressing these challenges while leveraging local strengths can contribute to a sustainable, well-integrated surveillance system that improves maternal and child health outcomes.

## Supporting information

**S1 Table. Characteristics of respondents for the in-depth interviews and focus group discussions [qualitative analysis].**
(DOCX)

**S2 Table. Quantitative analysis: Respondent's attitudes on maternal and child health.**
(DOCX)

**S3 Table. Quantitative Analysis: Monitoring of Mothers' and Children's Health (n = 601; 50% men).**
(DOCX)

**S4 Table. Quantitative analysis: Perceptions towards healthcare facilities (n = 601; 50% men).**
(DOCX)

**S5 Table. Quantitative analysis: Where to report when health issues are detected in mothers and children (n = 601; 50% men).**
(DOCX)

## Author contributions

**Conceptualization:** Dumilah Ayuningtyas, Asri C Adisasmita, Sabarinah Prasetyo.

**Data curation:** Fathimah S Sigit, Fitra Yelda, Dumilah Ayuningtyas, Asri C Adisasmita, Sabarinah Prasetyo.

**Formal analysis:** Fathimah S Sigit, Fitra Yelda, Dumilah Ayuningtyas, Asri C Adisasmita, Sabarinah Prasetyo.

**Funding acquisition:** Sabarinah Prasetyo.

**Investigation:** Fathimah S Sigit, Fitra Yelda, Dumilah Ayuningtyas, Asri C Adisasmita, Sabarinah Prasetyo.

**Methodology:** Fathimah S Sigit, Fitra Yelda, Dumilah Ayuningtyas, Asri C Adisasmita, Sabarinah Prasetyo.

**Project administration:** Fitra Yelda, Sabarinah Prasetyo.

**Resources:** Fitra Yelda, Dumilah Ayuningtyas, Asri C Adisasmita, Sabarinah Prasetyo.

**Software:** Fathimah S Sigit.

**Supervision:** Dumilah Ayuningtyas, Asri C Adisasmita, Sabarinah Prasetyo.

**Validation:** Fathimah S Sigit, Fitra Yelda, Dumilah Ayuningtyas, Asri C Adisasmita, Sabarinah Prasetyo.

**Visualization:** Fathimah S Sigit.

**Writing – original draft:** Fathimah S Sigit.

**Writing – review & editing:** Fitra Yelda, Dumilah Ayuningtyas, Asri C Adisasmita, Sabarinah Prasetyo.

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
