## [Decision Letter · Decision Letter 0]

11 Feb 2025

Dear Dr. Sigit,

Thank you for submitting your manuscript to PLOS ONE. After careful consideration, we feel that it has merit but does not fully meet PLOS ONE’s publication criteria as it currently stands. Therefore, we invite you to submit a revised version of the manuscript that addresses the points raised during the review process.

We look forward to receiving your revised manuscript.

Kind regards,

Ammal Mokhtar Metwally, Ph.D (MD)

Academic Editor

PLOS ONE

Journal Requirements:

 “This study is supported by the Indonesia Endowment Fund for Education (Indonesian: Lembaga Pengelola Dana Pendidikan; LPDP), Grant Number: NKB-696/UN2.RST/HKP.05.00/2021 PRIME (Partnership in Research Indonesia and Melbourne). The funder had no role in study design, data collection and analysis, decision to publish, or preparation of this manuscript.”

a) If there are ethical or legal restrictions on sharing a de-identified data set, please explain them in detail (e.g., data contain potentially identifying or sensitive patient information, data are owned by a third-party organization, etc.) and who has imposed them (e.g., a Research Ethics Committee or Institutional Review Board, etc.). Please also provide contact information for a data access committee, ethics committee, or other institutional body to which data requests may be sent

4. We notice that your supplementary tables are included in the manuscript file. Please remove them and upload them with the file type 'Supporting Information'. Please ensure that each Supporting Information file has a legend listed in the manuscript after the references list.

Reviewers' comments:

Reviewer's Responses to Questions

**Comments to the Author**

1. Is the manuscript technically sound, and do the data support the conclusions?

Reviewer #1: Yes

Reviewer #2: Yes

Reviewer #3: Yes

2. Has the statistical analysis been performed appropriately and rigorously?

Reviewer #1: Yes

Reviewer #2: Yes

Reviewer #3: Yes

3. Have the authors made all data underlying the findings in their manuscript fully available?

Reviewer #1: Yes

Reviewer #2: Yes

Reviewer #3: Yes

4. Is the manuscript presented in an intelligible fashion and written in standard English?

Reviewer #1: Yes

Reviewer #2: Yes

Reviewer #3: Yes

Reviewer #1: The manuscript is neat and well prepared, I would have only suggested that the discussion may need to be summarized little bit as there are many details from the results are repeated here, it may be better to point out the important findings and dirrectly connect that with other studies.

Some sentences like this "We observed that around 15.1%-24.1% of the general public appeared to have insufficient knowledge" need to be corrected, instead of "general public" it is suitable to say "study population". At the abstract you mentioned " Men exhibited higher ignorance rates (34.2%, 35.2%, 23.2%) than women (10.7%, 12.9%, 7.0%).", I would suggest that the you change the word "ignorance" as this difinition is broad and cannot be determined subject wise. May be you could use "less knowledgable to the subject matter..."

Reviewer #2: the study explores the effectiveness of community based surveillance system for maternal and child health in Depok, Indonesia. The methods used in the study appear to be appropriate and well suited for the research objectives. The mixed-method approach allows for the collection of broad statistical data and in-depth insights from stakeholders. The findings are beneficial for the improvement of the already established community health services in Indonesia.

Reviewer #3: Title - Barriers and Opportunities in Developing Community-Based Maternal and Child Health

Surveillance: A Mixed Methods Study in Depok, Indonesia

The title is appropriate

Abstract – Abstract is well structured but the section sub-titled as “Discussion” should be changed to “Conclusion”. Key words provided

Introduction – Statement of problem is well written. The magnitude of the problem, rationale for the study and study objectives are well presented.

Methods – The specific name of the study design is not mentioned. What is mentioned is the approach. This appears to be a descriptive cross-sectional study with a mixed methods approach.

• Study setting is well described.

• The minimum sample size estimation for the quantitative part of the study not calculated. appropriately written.

• What type of random selection was used to select the households?

• Data collection and analysis are well written

• Provide information on the number of interviews and FGDs conducted.

Result – the findings are fairly well presented.

• …….. disproportionately higher in men (34.2%) than women (10.7%). - Is this difference found to be statistically significant? This should be analysed

• ……… disproportionate proportion between men (35.2%) and women (12.9%). - Is this difference found to be statistically significant? This should be analysed

• The qualitative findings should be improved upon by inserting some appropriate quotes in the various sub-themes even though this is on table 3.

Discussion – Findings fairly well discussed.

There is no conclusion section after the discussion. The study limitations were not mentioned.

References - adequate

**Do you want your identity to be public for this peer review?** For information about this choice, including consent withdrawal, please see our Privacy Policy

Reviewer #1: No

Reviewer #2: **Yes: ** Rukhsana Ahmed, MD, PhD; Epidemiologist

Reviewer #3: **Yes: ** Prof. Tanimola Makanjuola Akande

---

## [Author Response · Author response to Decision Letter 1]

10 Apr 2025

We thank the three reviewers for the overall positive feedback and support of our manuscript, and the editors for the opportunity to submit an improved manuscript. We are grateful for the insightful comments and constructive suggestions, which have helped us improve the quality of our work. Please find below our point-by-point response addressing the feedback from the reviewers.

Reviewer's Responses to Questions

1. Is the manuscript technically sound, and do the data support the conclusions? The manuscript must describe a technically sound piece of scientific research with data that supports the conclusions. Experiments must have been conducted rigorously, with appropriate controls, replication, and sample sizes. The conclusions must be drawn appropriately based on the data presented.

Reviewer #1: Yes

Reviewer #2: Yes

Reviewer #3: Yes

2. Has the statistical analysis been performed appropriately and rigorously?

Reviewer #1: Yes

Reviewer #2: Yes

Reviewer #3: Yes

3. Have the authors made all data underlying the findings in their manuscript fully available? The PLOS Data policy requires authors to make all data underlying the findings described in their manuscript fully available without restriction, with rare exception (please refer to the Data Availability Statement in the manuscript PDF file). The data should be provided as part of the manuscript or its supporting information, or deposited to a public repository. For example, in addition to summary statistics, the data points behind means, medians and variance measures should be available. If there are restrictions on publicly sharing data—e.g., participant privacy or use of data from a third party—those must be specified.

Reviewer #1: Yes

Reviewer #2: Yes

Reviewer #3: Yes

4. Is the manuscript presented in an intelligible fashion and written in standard English? PLOS ONE does not copyedit accepted manuscripts, so the language in submitted articles must be clear, correct, and unambiguous. Any typographical or grammatical errors should be corrected at revision, so please note any specific errors here.

Reviewer #1: Yes

Reviewer #2: Yes

Reviewer #3: Yes

5. Review Comments to the Author

Reviewer #1: The manuscript is neat and well prepared, I would have only suggested that the discussion may need to be summarized little bit as there are many details from the results are repeated here, it may be better to point out the important findings and directly connect that with other studies. Some sentences like this "We observed that around 15.1%-24.1% of the general public appeared to have insufficient knowledge" need to be corrected, instead of "general public" it is suitable to say "study population". At the abstract you mentioned "Men exhibited higher ignorance rates (34.2%, 35.2%, 23.2%) than women (10.7%,12.9%, 7.0%).", I would suggest that the you change the word "ignorance" as this definition is broad and cannot be determined subject wise. Maybe you could use "less knowledgeable to the subject matter..."

We thank Reviewer #1 for their constructive feedback on our manuscript. In response to the suggestion, we have summarized the Discussion while retaining its most important information. Additionally, we created a Conclusion section at the end of the Discussion to highlight the main findings and propose our recommendation to the field. Line 429 in the revised manuscript, highlighted in yellow

Line 347 in the revised manuscript, highlighted in yellow

For the sentence "We observed that around 15.1%-24.1% of the general public appeared to have insufficient knowledge of …", we have now amended the sentence into "We observed that around 15.1%-24.1% of the study population appeared to have insufficient knowledge of …"

Line 41 in the revised manuscript, highlighted in yellow

For the sentence “Men exhibited higher ignorance rates (34.2%, 35.2%, 23.2%) than women (10.7%,12.9%, 7.0%)”, we have also revised the sentence into “Men appeared to be less knowledgeable on these matters (34.2%, 35.2%, 23.2%) than women (10.7%,12.9%, 7.0%)." in accordance with the reviewer's recommendation.

Reviewer #2: The study explores the effectiveness of community-based surveillance system for maternal and child health in Depok, Indonesia. The methods used in the study appear to be appropriate and well suited for the research objectives. The mixed-method approach allows for the collection of broad statistical data and in-depth insights from stakeholders. The findings are beneficial for the improvement of the already established community health services in Indonesia.

Minor issues: I suggest that the authors provide in a table format or otherwise some characteristics of the various respondents chosen for the in-depth interviews such as their age group, years of service in their respective positions. For example, break down the years of service to less than 3 years, between 3-5 years, 5-10 years, and greater than 10 years. This information would add value to the information provided. In addition, a description of the composition of the participants in the FGDs and their age range, how many men and women participated in the FGDs. It would enable the readers to get an understanding of the value of the data collected.

We deeply thank and appreciate Reviewer #2 for his/her overall positive response and support of our manuscript. To address the suggestion, we have now provided more details on the characteristics of the respondents or participants for the interviews and FGDs.

Line 165 in the revised manuscript, highlighted in yellow

“The characteristics of the respondents for the in-depth interviews and FGDs were described in Supplemental Table 1.”

(We renumbered the supplemental tables, as this table appears first in the order of the manuscript).

Supplemental Table 1: Characteristics of Respondents for the In-depth Interviews and Focus Group Discussions [Qualitative Analysis]

Discussion Section:

Elaborate on the key findings. Highlight the most important findings prominently. For instance, the disparity in knowledge between men and women and high approval rate of monitoring could be emphasized.

Recommendations: Offer clear and actionable recommendations based on your findings. For example, the authors could suggest targeted public health education programs for men.

Conclusion: summarize the key points and emphasize the importance of this study.

We have now incorporated these suggestions to enrich our Discussion, by creating a Conclusion section that highlights the key findings prominently and includes actionable recommendations for future implementation.

Line 429 in the revised manuscript, highlighted in yellow

Conclusion

“This study highlights the barriers and opportunities in establishing a community-based surveillance system for maternal and child health in Depok, Indonesia. We observed that while most community members support health monitoring, knowledge gaps—particularly among men—persist, and logistical challenges such as high population mobility, health volunteer capacity limitations, and social stigma hinder effective implementation. However, strong community engagement, existing empowerment programs, digital communication channels, and stakeholder collaboration offer promising opportunities. Addressing these challenges while leveraging local strengths can contribute to a sustainable, well-integrated surveillance system that improves maternal and child health outcomes. Future efforts should focus on improving public health education, particularly for men to encourage their involvement in ensuring the health of mothers and children, enhancing health volunteer training, and strengthening intersectoral coordination to ensure long-term success.”

Other points:

1. I found that the authors have not described any limitations they had in planning and doing the study. I suggest providing a paragraph on what the authors feel might be the limitations of this study would strengthen the manuscript.

2. In supplement table-2, I suggest instead of the options “very agree” and “very disagree”, these be stated as “fully agree” and “fully disagree”

We have now mentioned several strengths and limitations of the study, and renamed the options in Supplemental Table 2 as suggested.

Line 414 in the revised manuscript, highlighted in yellow

Strengths and Limitations of the Study

“This study employs a robust mixed-methods approach, integrating quantitative surveys with qualitative insights to provide a comprehensive understanding of community-based maternal and child health surveillance. The large sample size (300 households) and diverse stakeholder insights enhance representativeness and real-world applicability, capturing on-the-ground challenges and best practices from multiple levels of community and healthcare stakeholders, making the findings highly relevant for policy and intervention planning.”

“However, several limitations exist. The cross-sectional design restricts the ability to infer causal relationships between identified factors and surveillance effectiveness. Additionally, self-reported data in the household survey may be subject to recall or social desirability bias, particularly in sensitive topics such as stigma and healthcare-seeking behaviors. Moreover, while the study covers three subdistricts in Depok, its generalizability to other regions in Indonesia may be limited due to variations in health system infrastructure and socio-cultural factors. Future research should consider longitudinal studies and broader geographic coverage to strengthen applicability.”

Supplemental Table 2 in the revised manuscript, highlighted in yellow

Reviewer #3: Title - Barriers and Opportunities in Developing Community-Based Maternal and Child Health Surveillance: A Mixed Methods Study in Depok, Indonesia. The title is appropriate.

Abstract – Abstract is well structured but the section sub-titled as “Discussion” should be changed to “Conclusion”. Key words provided.

We thank Reviewer #3 for his/her valuable feedback, which have helped clarify and strengthen our manuscript. In the Abstract, we have changed the subtitle "Discussion" to "Conclusion" as recommended. Line 50 in the revised manuscript, highlighted in yellow

Introduction – Statement of problem is well written. The magnitude of the problem, rationale for the study and study objectives are well presented.

Methods – The specific name of the study design is not mentioned. What is mentioned is the approach. This appears to be a descriptive cross-sectional study with a mixed methods approach. • Study setting is well described. • The minimum sample size estimation for the quantitative part of the study not calculated. appropriately written. • What type of random selection was used to select the households? • Data collection and analysis are well written • Provide information on the number of interviews and FGDs conducted.

• We have now stated the study design as “a descriptive cross-sectional study with a mixed methods approach” to ensure clarity. Line 124 in the revised manuscript, highlighted in yellow

• We have now included information on the minimum sample size estimation of the quantitative study, as suggested.

Line 134 in the revised manuscript, highlighted in yellow

“Quantitative data were collected from 300 households residing in three subdistricts in Depok (Mekarjaya, Curug, and Tirtajaya) with an interviewer-assisted questionnaire. The three subdistricts were purposively selected by the Depok City Health Office, based on their relatively higher prevalence of maternal and children health issues compared to other subdistricts. The sample size was 100 households per subdistrict, calculated based on a 95% confidence level, a 10% precision level, a 50% proportion of events, and a design effect of 1.”

• We have provided additional details on the sampling procedures for selecting the households.

Line 140 in the revised manuscript, highlighted in yellow

“The sampling procedure followed a two-stage cluster and random sampling method within each subdistrict to select the households. In the first stage, twenty neighbourhoods/hamlets from each of the three subdistricts were selected as the clusters using the probability proportional to size method. In the second stage, five households per hamlet were randomly selected using a computer-based random generator from the household list of each selected hamlet.”

• We have specified the number of interviews and FGDs conducted throughout the study, along with the respondents, stakeholders, and community representatives involved in each session (detailed in Supplemental Table 1, as in our earlier response to Reviewer #2).

Result – the findings are fairly well presented.

• …….. disproportionately higher in men (34.2%) than women (10.7%). - Is this difference found to be statistically significant? This should be analysed.

• ……… disproportionate proportion between men (35.2%) and women (12.9%). - Is this difference found to be statistically significant? This should be analysed.

The differences were statistically significant. We have incorporated the p-values for differences in knowledge of pregnancy, childbirth, postpartum, and newborn care between men and women in Table 2.

• The qualitative findings should be improved upon by inserting some appropriate quotes in the various sub-themes even though this is on table 3.

We have incorporated several direct respondent quotes from Table 3 into the Results section.

Line 233, 245, 252, 258, 264, 277, 288, 299, 318, 336 in the revised manuscript, highlighted in yellow

Discussion – Findings fairly well discussed. There is no conclusion section after the discussion. The study limitations were not mentioned.

References - adequate

We have now outlined several limitations of our study, as well as created a Conclusion section within the Discussion (as pointed out earlier in our response to Reviewer #2)

Line 414 in the revised manuscript, highlighted in yellow (Study Strengths & Limitations)

Line 429 in the revised manuscript, highlighted in yellow (Conclusion)

---

## [Decision Letter · Decision Letter 1]

29 May 2025

Dear Dr. Sigit,

Thank you for submitting your manuscript to PLOS ONE. After careful consideration, we feel that it has merit but does not fully meet PLOS ONE’s publication criteria as it currently stands. Therefore, we invite you to submit a revised version of the manuscript that addresses the points raised during the review process.

We look forward to receiving your revised manuscript.

Kind regards,

Ammal Mokhtar Metwally, Ph.D (MD)

Academic Editor

PLOS ONE

Journal Requirements:

Additional Editor Comments:

Thank you for your resubmission of the manuscript entitled “Barriers and Opportunities in Developing Community-Based Maternal and Child Health Surveillance: A Mixed Methods Study in Depok, Indonesia.” We appreciate the thoughtful and comprehensive revisions you have made in response to reviewer feedback.

Upon further editorial review, we find your manuscript significantly improved and of potential value to public health professionals, particularly in resource-limited contexts. The integration of both quantitative and qualitative data, along with an emphasis on real-world application, is commendable. However, a few remaining issues must be addressed before final acceptance.

1. Clarify and streamline the abstract

• The abstract is informative but could benefit from improved clarity and brevity.

• Certain terms such as “contrariwise” are less common in scientific writing and may hinder accessibility.

• Replace “contrariwise” with “alternatively” and consider condensing long sentences for improved readability.

2. Refine discussion of results

• The discussion section contains repetition of results. While referencing findings is useful, repeating detailed data limits space for deeper interpretation and policy relevance.

• Instead of repeating knowledge percentages from the Results, use this space to propose how these gaps should inform health education initiatives.

3. Deepen policy implications in the conclusion

• The conclusion provides practical recommendations, but could be more action-oriented. A well-articulated roadmap strengthens the utility of the findings for policymakers and health system planners.

• Suggest launching a pilot community surveillance initiative integrating digital tools, with evaluation benchmarks based on this study.

4. Enhance transparency of qualitative insights

• Although direct quotes have been added, representation of voices could be more balanced. Quotes are largely from officials or volunteers, with limited insights from family/community members.

• Consider incorporating more quotes from mothers or household heads to reflect end-user perspectives.

5. Language and terminology refinement

• Some phrases may benefit from more precise or neutral terminology. Terms like “ignorance rates” (now corrected) and “exclusive neighborhoods” could be rephrased for neutrality and clarity.

• Use “less knowledgeable” instead of “ignorant,” and “gated communities” or “higher-income areas” instead of “exclusive neighborhoods.”

6. Reference formatting and integration

• Citations are appropriate, but integration in the narrative could be smoother. Over reliance on parenthetical citation disrupts narrative flow.

• Use phrases such as “as shown by Sharma et al. (2023)” instead of placing multiple bracketed references at the sentence end.

7. Supplemental material referencing

• Supplemental tables provide valuable data but are not always clearly referenced in the main text. Cross-referencing enhances reader navigation and data transparency.

• Add phrases like “(see Supplemental Table 2)” directly after discussing survey items.

Final Comments:

Your manuscript has undergone meaningful improvement in response to previous reviewer suggestions. The addition of statistical significance tests, a dedicated conclusion, and clearer methodological explanations have substantially improved the manuscript’s quality. With minor revisions primarily around clarity, structure, and terminology, the manuscript will be well-positioned for publication.

We appreciate your contribution to advancing maternal and child health systems, especially in underserved settings.

Reviewers' comments:

Reviewer's Responses to Questions

**Comments to the Author**

Reviewer #1: (No Response)

Reviewer #2: All comments have been addressed

2. Is the manuscript technically sound, and do the data support the conclusions?

Reviewer #1: Yes

Reviewer #2: Yes

3. Has the statistical analysis been performed appropriately and rigorously?

Reviewer #1: Yes

Reviewer #2: Yes

4. Have the authors made all data underlying the findings in their manuscript fully available?

Reviewer #1: Yes

Reviewer #2: Yes

5. Is the manuscript presented in an intelligible fashion and written in standard English?

Reviewer #1: Yes

Reviewer #2: Yes

Reviewer #1: The authors have satisfactory addressed most of my previous recomendations, but I am not still confissed the flow of the discussion part. You don't have to mention study aim again in here to start with, at the disscussion just mention your main findings, first finding and you interpretation flowed by comparing other studies.

I would sugest to flow a clear line of presentation:

1.Your study results, two to three main findings

2. You can compare other studies to each point or as you did now mention all findings and then comparing the other studies to the equavalent finding.

3. Mention your limitations here.

3. Make your recommendations here or in the conclussion part at last section, not in the middle of the discussion as you mentioned in L366.

Reviewer #2: (No Response)

**Do you want your identity to be public for this peer review?** For information about this choice, including consent withdrawal, please see our Privacy Policy

Reviewer #1: **Yes: ** Abdulkadir Ismael Ahmed

Reviewer #2: No

---

## [Author Response · Author response to Decision Letter 2]

28 Jul 2025

We thank the Academic Editor and Reviewers for their careful review and thorough feedback on our manuscript. We found the detailed comments to be constructive and have revised the manuscript accordingly. Please find below our point-by-point responses addressing each comment from the Editor and Reviewers.

Additional Editor Comments:

Thank you for your resubmission of the manuscript entitled “Barriers and Opportunities in Developing Community-Based Maternal and Child Health Surveillance: A Mixed Methods Study in Depok, Indonesia.” We appreciate the thoughtful and comprehensive revisions you have made in response to reviewer feedback.

Upon further editorial review, we find your manuscript significantly improved and of potential value to public health professionals, particularly in resource-limited contexts. The integration of both quantitative and qualitative data, along with an emphasis on real-world application, is commendable. However, a few remaining issues must be addressed before final acceptance.

1. Clarify and streamline the abstract

• The abstract is informative but could benefit from improved clarity and brevity. Certain terms such as “contrariwise” are less common in scientific writing and may hinder accessibility. Replace “contrariwise” with “alternatively” and consider condensing long sentences for improved readability.

We have replaced “contrariwise” with “alternatively” and revised several long sentences to improve clarity and conciseness.

Lines 45-46, highlighted in yellow in the manuscript

“… and reluctance among families with middle-high socioeconomic status. Alternatively, opportunities included positive community perceptions …”.

2. Refine discussion of results

• The discussion section contains repetition of results. While referencing findings is useful, repeating detailed data limits space for deeper interpretation and policy relevance.

• Instead of repeating knowledge percentages from the Results, use this space to propose how these gaps should inform health education initiatives.

We have removed all repeated mentions of the results and instead added a deeper interpretation and discussion of the policy relevance of our findings. Additionally, we have created a dedicated Public Health Implications section to propose recommendations based on our study results (see below).

3. Deepen policy implications in the conclusion

• The conclusion provides practical recommendations, but could be more action-oriented. A well-articulated roadmap strengthens the utility of the findings for policymakers and health system planners.

• Suggest launching a pilot community surveillance initiative integrating digital tools, with evaluation benchmarks based on this study.

We have now recommended actionable strategies based on our findings to strengthen maternal and child health surveillance and service delivery, including the pilot community-based surveillance initiative suggested by the editor. These recommendations are compiled in a dedicated 'Public Health Implications' section at the end of the Discussion.

Line 449, highlighted in yellow in the manuscript

Public Health Implications

This study highlights several actionable strategies to strengthen maternal and child health surveillance and service delivery. First, targeted maternal and child health education, particularly for men on obstetric danger signs, alongside efforts to foster paternal involvement, should be promoted, as husbands play a crucial role in household decision-making and access to healthcare services. Second, to prevent under-surveillance problems among migrant populations, gated communities, and stigmatized cases, tailored approaches could work, including: (1) proactive outreach in densely populated rental housing clusters and informal settlements to better reach migrant households; (2) cooperation with resident associations and private clinics to improve surveillance participation among higher socioeconomic groups in gated communities; and (3) public campaigns to reduce stigma and normalize health-seeking behaviour to better to capture sensitive cases, e.g., unwanted pregnancies. Third, leveraging user-friendly digital tools such as mobile-phone-based data reporting systems may help increase participation in community-based surveillance, including in the three aforementioned groups. A pilot initiative integrating digital tools into community-based surveillance is recommended as the next step, with this study serving as a benchmark for future evaluations. Lastly, strengthening the role, capacity, and support for health volunteers is essential, and this can be achieved by providing regular training and basic equipment, genuine recognition, and small incentives (transport allowances) to sustain volunteer performance. These strategies can be implemented through intersectoral collaboration across the health, social welfare, education, and civil registration sectors.

4. Enhance transparency of qualitative insights

• Although direct quotes have been added, representation of voices could be more balanced. Quotes are largely from officials or volunteers, with limited insights from family/community members. Consider incorporating more quotes from mothers or household heads to reflect end-user perspectives.

We had already included community members in our qualitative analysis, as shown in Table 3 and Supplementary Table 1. All three community member representatives interviewed were women, and most of them (2 out of 3) had lived in the neighborhoods for more than 10 years. We considered length of residence in our selection due to the city’s high migration rates, which may limit respondents’ familiarity with ongoing community issues.

Community insights were further strengthened by including heads of neighborhoods or hamlets, who provided broader community perspectives (also shown in Table 3 and Supplementary Table 1). These individuals were appointed by the community to lead neighborhood activities. As they are not paid by the government or receive any formal remuneration, their neutrality is expected.

Nevertheless, we have added a suggestion for future studies to directly interview pregnant women, new mothers, and men or husbands as heads of households to better capture the end-user perspective, as in our study, data from these groups were collected only through a quantitative survey.

Lines 442, highlighted in yellow in the manuscript

“Although the quantitative survey included community members (heads of households) involved in community-based surveillance, the qualitative analysis has yet to incorporate perspectives from pregnant women, mothers with infants, or men as husbands. Future studies are encouraged to include these groups in qualitative interviews to better capture end-user perspectives.”

5. Language and terminology refinement

• Some phrases may benefit from more precise or neutral terminology. Terms like “ignorance rates” (now corrected) and “exclusive neighborhoods” could be rephrased for neutrality and clarity. Use “less knowledgeable” instead of “ignorant,” and “gated communities” or “higher-income areas” instead of “exclusive neighborhoods.”

We have replaced “exclusive neighborhoods” with “gated communities” in favor of a more objective term.

Lines 264, 389, 454, 458, and Table 3, highlighted in yellow in the manuscript

“There is a challenge in reaching the residents who live in gated communities, as they are generally very difficult to be visited.”

“Interestingly, we observed that households with higher socioeconomic status, or those residing in gated communities, were …”

“Second, to prevent under-surveillance problems among migrant populations, gated communities, and stigmatized cases, tailored approaches could work, …”

“… and private clinics to improve surveillance participation among higher socioeconomic groups in gated communities;”

“Residents with middle-high income socioeconomic status who live in gated communities are …”

6. Reference formatting and integration

• Citations are appropriate, but integration in the narrative could be smoother. Over reliance on parenthetical citation disrupts narrative flow. • Use phrases such as “as shown by Sharma et al. (2023)” instead of placing multiple bracketed references at the sentence end.

• Supplemental tables provide valuable data but are not always clearly referenced in the main text. Cross-referencing enhances reader navigation and data transparency. • Add phrases like “(see Supplemental Table 2)” directly after discussing survey items.

We have now integrated the citations into the narrative and ensured that the results presented in the main text are clearly referenced to the corresponding tables or supplementary materials.

Lines 357, 362, 371, 378, 385, 393, 399, 402, 409, 420, 427, highlighted in yellow in the manuscript

“…, our observation that men had less knowledge of warning signs in pregnancy, childbirth, and newborn care was consistent with a study from Mersha (2019), which reported that men were often less aware …”

“…, Rahman et al. (2018) found that pregnant women who were accompanied by their husbands …”

“Similarly, Kebede et al. (2022) also showed that low paternal involvement …”

“Rogeaux (2022) found that migrant children in urban areas had lower weight gain, ... Sharma (2023) attributed this to financial hardship, inadequate feeding practices, … Khan (2020) further noted that ‘urban poor’ populations with unfavorable social determinants of health (SDoH) are …”

“…, Kim (2019) reported that stigma towards unmarried mothers caused them to hide their pregnancy and avoided antenatal care. Mukaliburt (2022) similarly observed that stigmatization toward …”

“Czeisler (2020) noted that approximately 12% of routine medical care, …, were delayed during the pandemic”

“For example, Roberts et al. (2020) and Jang (2019) reported that in the United States, survey response rates were particularly low among ... Van Loon (2003) highlighted that such non-response can bias prevalence estimates, …”

“Diese (2018) observed that mobile phones were widely perceived by the community as …. “

“Verity et al. (2022) emphasized that timely detection of disease through surveillance can lead to …”

“In Thailand, for example, Kaweenuttayanon et al. (2021) reported that during the COVID-19 pandemic, …”

“These successes were possible, as noted by Sakeah et al. (2021), largely driven by the health volunteers' ability to …”

“Nontapet (2022) and Owusu (2023) reported that community health workers perceived regular training as ...”

“Yet, as noted by Bezbaruah (2021), such training is not consistently provided.”

“Sunguya (2017) and Bezbaruah (2021) also highlighted that the lack of essential equipment, …”

“As also suggested by Sunguya (2017), Nontapet (2022), and Sakeah (2021), if the budget allows, small monetary incentives …”

Lines 165, 182, 192, 200, 208, 211, 215, 219, 222, 232, 268, highlighted in yellow in the manuscript

“The characteristics of the respondents for the in-depth interviews and FGDs were described in Supplemental Table 1.”

“As shown in Table 1, the quantitative study population has …”

“… was disproportionately higher in men (34.2%) than women (10.7%) (see Table 2).”

“…, and this is more so in male (23.2%) than female (7.0%) respondents (see Table 2).”

“…, and 59.9% disagree if childbirth is assisted by non-medical professionals (e.g., shamans) (see Supplemental Table 2).”

“…, the majority of (98%) respondents said that they would approve such monitoring to be conducted (see Supplemental Table 3).”

“The majority (>82%) also did not have problems obtaining healthcare services (see Supplemental Table 4).”

“…, health professionals (11.6-12.9%), and community leaders (11.0%-14.1%) (see Supplemental Table 5).”

“Qualitative Analysis: Barriers to Establishing a Community-Based Surveillance System for Maternal and Child Health [Table 3]”

“…, as illustrated in the following direct quotes. Complete statements are available in Table 3.”

Final Comments: Your manuscript has undergone meaningful improvement in response to previous reviewer suggestions. The addition of statistical significance tests, a dedicated conclusion, and clearer methodological explanations have substantially improved the manuscript’s quality. With minor revisions primarily around clarity, structure, and terminology, the manuscript will be well-positioned for publication. We appreciate your contribution to advancing maternal and child health systems, especially in underserved settings.

We thank the Academic Editor for their overall support of our manuscript.

Reviewer's Comments and Responses to Questions:

1. If the authors have adequately addressed your comments raised in a previous round of review and you feel that this manuscript is now acceptable for publication, you may indicate that here to bypass the “Comments to the Author” section, enter your conflict of interest statement in the “Confidential to Editor” section, and submit your "Accept" recommendation.

Reviewer #1: (No Response)

Reviewer #2: All comments have been addressed

2. Is the manuscript technically sound, and do the data support the conclusions? The manuscript must describe a technically sound piece of scientific research with data that supports the conclusions. Experiments must have been conducted rigorously, with appropriate controls, replication, and sample sizes. The conclusions must be drawn appropriately based on the data presented.

Reviewer #1: Yes

Reviewer #2: Yes

3. Has the statistical analysis been performed appropriately and rigorously?

Reviewer #1: Yes

Reviewer #2: Yes

4. Have the authors made all data underlying the findings in their manuscript fully available? The PLOS Data policy requires authors to make all data underlying the findings described in their manuscript fully available without restriction, with rare exception (please refer to the Data Availability Statement in the manuscript PDF file). The data should be provided as part of the manuscript or its supporting information, or deposited to a public repository. For example, in addition to summary statistics, the data points behind means, medians and variance measures should be available. If there are restrictions on publicly sharing data—e.g. participant privacy or use of data from a third party—those must be specified.

Reviewer #1: Yes

Reviewer #2: Yes

5. Is the manuscript presented in an intelligible fashion and written in standard English? PLOS ONE does not copyedit accepted manuscripts, so the language in submitted articles must be clear, correct, and unambiguous. Any typographical or grammatical errors should be corrected at revision, so please note any specific errors here.

Reviewer #1: Yes

Reviewer #2: Yes

6. Review Comments to the Author

Reviewer #1: The authors have satisfactory addressed most of my previous recommendations, but I am not still convinced the flow of the discussion part. You don't have to mention study aim again in here to start with, at the discussion just mention your main findings, first finding and you interpretation flowed by comparing other studies. I would suggest to flow a clear line of presentation:

1.Your study results, two to three main findings 2. You can compare other studies to each point or as you did now mention all findings and then comparing the other studies to the equivalent finding.

3. Mention your limitations here. 4. Make your recommendations here or in the conclusion part at last section, not in the middle of the discussion as you mentioned in L366.

We thank Reviewer #1 for their thorough and constructive feedback on our manuscript. In response, we have revised the Discussion section to improve its logical flow and coherence. Following the reviewer’s advice, we now begin the section by stating ou

---

## [Decision Letter · Decision Letter 2]

1 Sep 2025

Barriers and Opportunities in Developing Community-Based Maternal and Child Health Surveillance: A Mixed Methods Study in Depok, Indonesia

PONE-D-24-59311R2

Dear Dr. Sigit,

We’re pleased to inform you that your manuscript has been judged scientifically suitable for publication and will be formally accepted for publication once it meets all outstanding technical requirements.

Kind regards,

Ammal Mokhtar Metwally, Ph.D (MD)

Academic Editor

PLOS ONE

Additional Editor Comments (optional):

Reviewer #1:

Reviewers' comments:

Reviewer's Responses to Questions

**Comments to the Author**

Reviewer #1: All comments have been addressed

2. Is the manuscript technically sound, and do the data support the conclusions?

Reviewer #1: Yes

3. Has the statistical analysis been performed appropriately and rigorously?

Reviewer #1: Yes

4. Have the authors made all data underlying the findings in their manuscript fully available?

Reviewer #1: Yes

5. Is the manuscript presented in an intelligible fashion and written in standard English?

Reviewer #1: Yes

Reviewer #1: The authors have addressed all the comments made in the previous reviews, and I can confidently say that this article is nearly complete, requiring only a few minor adjustments. First, out of the 10 quotes mentioned in the results, only one is from community members. Since you have the data, I suggest incorporating more quotes from the community to enhance the representation.

The discussion section has improved significantly but still feels a bit bulky due to repetitive sentences and paragraphs. I believe the discussion text could be trimmed by half while maintaining the content, making it more concise and easier to read. For instance, instead of listing all findings in a long paragraph (like in line 348), you could directly present your first finding and then transition with statements like, "This is consistent with..." or "In contrast, another study..."

Additionally, the phrase "Comparing our results with other studies" is used repeatedly, which isn't necessary. You could simply connect your findings to other studies without restating the comparison.

There is a minor grammatical error in line 376, but it should be resolved with the final adjustments. Also, in line 366, consider connecting the sentences, as the first sentence currently stands alone and requires its own reference.

**Do you want your identity to be public for this peer review?** For information about this choice, including consent withdrawal, please see our Privacy Policy

Reviewer #1: **Yes: ** Abdulkadir Ismael Ahmed

---

## [Editor Report · Acceptance letter]

PONE-D-24-59311R2

PLOS ONE

Dear Dr. Sigit,

I'm pleased to inform you that your manuscript has been deemed suitable for publication in PLOS ONE. Congratulations! Your manuscript is now being handed over to our production team.

Kind regards,

on behalf of

Professor Ammal Mokhtar Metwally

Academic Editor

PLOS ONE